# ST3DNetCrime: Improved ST-3DNet Model for Crime Prediction at Fine Spatial Temporal Scales

Qifen Dong [1,2,*], Yu Li [3], Ziwan Zheng [2,4], Xun Wang [1] and Guojun Li [5]

1   Department of Computer and Information Security, Zhejiang Police College, Hangzhou 310053, China
2   Ministry of Public Security's Key Laboratory of Public Security Information Application Based on Big-Data Architecture, Hangzhou 310053, China
3   Hahang Network Technology Co., Ltd., Hangzhou 310012, China
4   School of Big-Data and Network Security, Zhejiang Police College, Hangzhou 310053, China
5   Department of Basic Courses, Zhejiang Police College, Hangzhou 310053, China
*   Correspondence: dongqifen@zjjcxy.cn

**Abstract:** Crime prediction is crucial for sustainable urban development and protecting citizens' quality of life. However, there exist some challenges in this regard. First, the spatio-temporal correlations in crime data are relatively complex and are heterogenous in time and space, hence it is difficult to model the spatio-temporal correlation in crime data adequately. Second, crime prediction at fine spatial temporal scales can be applied to micro patrol command; however, crime data are sparse in both time and space, making crime prediction very challenging. To overcome these challenges, based on the deep spatio-temporal 3D convolutional neural networks (ST-3DNet), we devise an improved ST-3DNet framework for crime prediction at fine spatial temporal scales (ST3DNetCrime). The framework utilizes diurnal periodic integral mapping to solve the problem of sparse and irregular crime data at fine spatial temporal scales. ST3DNetCrime can, respectively, capture the spatio-temporal correlations of recent crime data, near historical crime data and distant historical crime data as well as describe the difference in the correlations' contributions in space. Extensive experiments on real-world datasets from Los Angeles demonstrated that the proposed ST3DNetCrime framework has better prediction performance and enhanced robustness compared with baseline methods. In additon, we verify that each component of ST3DNetCrime is helpful in improving prediction performance.

**Keywords:** crime prediction; fine spatial temporal scales; deep learning; spatio-temporal features

## 1. Introduction

Crime prediction assists police departments and government authorities to formulate crime prevention strategies, and bears an important impact on urban sustainable development and citizens' quality of life [1]. Thus, researchers in industry and academia are actively studying this problem from various perspectives. In this paper, our goal is to infer the crime density of each spatial unit in the next time period based on historical crime records, which can provide guidance in terms of optimizing the arrangement of police patrols. This topic continues to increasingly attract the attention of researchers.

According to classical criminal theories such as routine activity theory [2], Near Repeat theory [3] and rational choice theory [4], the occurrence of crime is closely related to time and space. In the earlier stage, the crime prediction techniques primarily focused on either the temporal dimension of crime [5–7] or the spatial dimension of crime [8–10]. If both the temporal and spatial correlations in crime are considered simultaneously, it is expected that crime analysis and prediction research will be advanced in meaningful ways [11]. Therefore, with the development of spatiotemporal analysis technology, researchers are actively studying crime prediction using spatio-temporal methods in recent years [12]. However, some challenges abound:

Challenge 1 (Modeling the spatio-temporal correlation in crime data adequately).

On one hand, the crime within a region is influenced by the recent and distant historical crime of itself as well as its nearby or distant regions. In other words, the spatio-temporal correlations in crime data are relatively complex and difficult to extract effectively. On the other hand, the contributions of the correlations in crime data are different in time and space. Hence, it is necessary to consider the heterogeneity of the correlations' contribution in the crime prediction model, which is often ignored in most existing spatio-temporal crime prediction models.

Challenge 2 (Achieving crime prediction at fine spatial temporal scales).

In practice, predicting crime at fine spatial temporal scales can not only provide real-time basic intelligence for the daily patrol and investigation of police, but also provide quantitative basis for the optimal allocation of urban police resources, providing strong support for risk prevention and crime control. Hence, crime prediction at fine spatial temporal scales represents a significant scientific and practical issue [13]. However, crime data at fine spatial temporal scales are sparse in both time and space, making crime prediction a very challenging task, and there are not many research results so far.

To address the challenges, based on the deep spatio-temporal 3D convolutional neural networks (ST-3DNet) initially developed in [14] for traffic raster data prediction, we propose an improved ST-3DNet framework for crime prediction at fine spatial temporal scales (ST3DNetCrime). The following summarizes our main contributions.

- To the best of our knowledge, we are the first to introduce and improve the ST-3DNet model to make it suitable for crime prediction domain.
- Diurnal periodic integral mapping is used to solve the problem of sparse and irregular crime data at fine spatial temporal scales.
- We consider three categories of temporal properties of crime, i.e., closeness, period, and trend. Furthermore, we modify the ST-3DNet structure to, respectively, extract the three spatio-temporal correlations and describe the difference between the correlations' contributions in space.
- We conduct comprehensive experiments using real-world datasets gathered from Los Angeles to assess the performance of ST3DNetCrime model and examine the role of each component in the ST3DNetCrime model for crime prediction.

The remainder of this paper is organized as follows. Related works are summarized in Section 2. The Preliminaries, i.e., the definitions of crime prediction and ST-3DNet structure, are described in Section 3. Our proposed ST3DNetCrime model is expounded in Section 4. Section 5 discusses the experimental results. Finally, Section 6 provides conclusions and suggestions for future work.

## 2. Related Work

In this section, we mainly discuss related work on spatio-temporal crime prediction.

The spatio-temporal crime prediction models simultaneously consider both the temporal and spatial correlations in crime, and could provide great potential for the in-depth study on crime analysis. This topic has attracted increasing attention in recent years. For example, recent studies [15,16] presented a spatio-temporal Cokriging model to integrate historical crime data and environmental variables related to criminal patterns, such as urban transitional zones identified from nightlight imagery, and movement data of past offenders collected in routine police stop-and-question operations, for more accurate crime prediction. Zhao et al. validated the existence of temporal-spatial correlations in crime and developed a crime prediction method named TCP which models these correlations into a coherent framework [17]. Since the temporal dimension of crime is not considered in the popular kernel density estimation (KDE), a spatio-temporal kernel density estimation (STKDE) framework was proposed for predictive crime hotspot mapping and evaluation [18]. Considering that criminal behavior shares similarity with earthquakes, whereby the risk of subsequent earthquakes, or aftershocks, increases near the location of an initial event, self-exciting point process models in seismology were adapted to model the

crime [19]. Recently, Farjami and Abdi developed a genetic-fuzzy-based system which is a suitable tool to find spatio-temporal crime patterns and predict future crimes for environments with clustered crimes in space and time [20]. Deep learning has also recently been applied to the forecasting and modeling of crime. In [21], the problem of crime forecasting was expressed as a space-time series prediction problem, and an appropriate deep recurrent neural network with spatial influence embedding was implemented to predict criminal activity in the near future. By jointly embedding all spatial, temporal, and categorical signals into hidden representation vectors and capturing crime dynamics via an attentive hierarchical recurrent network, Huang et al. created a crime prediction model based on deep neural network architecture (DeepCrime) that can predict the occurrence of various crimes in each area of the city [22]. A Multi-View and Multi-Modal Spatial-Temporal learning framework (MiST) for the prediction of abnormal events in the city is studied in [23]. MiST can explicitly model the dynamic patterns of citywide abnormal events from spatial-temporal–categorical views, with the integration of a multi-modal pattern fusion module and a hierarchical recurrent framework. In addition, the authors of [24] proposed a deep temporal multi-graph convolutional network (DT-MGCN) model for crime prediction, which combines the spatial-temporal component and graph generation component to capture the relationships between crime and many external elements. However, the complex spatio-temporal correlations in crime data have not been adequately described. For instance, crime is not only highly correlated in adjacent time intervals, but also affected by other temporal properties, such as period patterns and trend patterns which are not efficiently captured in most existing spatio-temporal crime prediction models. Moreover, often neglected is that the spatio-temporal correlations in crime data are heterogenous in time and space. In addition, most of these techniques model the coarse-scale spatial-temporal patterns in the crime data.

Fortunately, new developments in deep learning techniques enable efficient modeling of the complex correlations in spatio-temporal data. Furthermore, there are research results in the field of traffic data prediction. For example, Zhang et al. [25,26] proposed a spatio-temporal residual network (ST-ResNet), which comprises an ensemble of deep residual networks [27] and convolution layers, to collectively predict hourly inflow and outflow of crowds in every region of a city. Subsequently, the ST-ResNet model is applied to crime prediction at fine spatial temporal scales. Wang et al. [13] adapted the ST-ResNet structure to predict the hourly crime distribution of Los Angeles in neighborhood-sized parcels. In addition, an adaptive spatial resolution method was proposed to select the best spatial resolution for hourly crime prediction based on the ST-ResNet model [28]. However, ST-ResNet only considers the information over neighboring time intervals as multiple channels, hence the input's temporal information is lost after the first convolution layer [14]. To overcome this limitation, Guo et al. [14] presented a spatio-temporal traffic forecasting network called ST-3DNet based on deep learning. ST-3DNet uses 3D convolutions and residual units to capture features from both spatial and temporal dimensions, and proposed a "Recalibration" (Rc) module to clearly state the contribution difference of the spatio-temporal correlations in space. However, ST-3DNet could not be applied to crime prediction at fine spatial temporal scales, because crime data are very sparse and bear more complex spatiotemporal dynamic relationships than traffic data. Hence, we draw upon the existing results on crime prediction at fine spatial temporal scales, and improve ST-3DNet to obtain better crime prediction performance.

## 3. Preliminaries

### 3.1. Problem Definition

The aim of this study is to predict crime at small spatial and hourly temporal scales. Here, the researched area is divided into 16 by 16 grid regions based on longitude and latitude. Let $X_t \in R^{16 \times 16}$ denote the crime matrix at the $t$-th time slot, where the element $(X_t)_{i,j}$ is the number of crime in grid $(i, j)$ at time slot $t$. The time interval between two time slots is one hour. Then the problem of crime prediction is described as follows.

**Definition 1.** *(Crime prediction). Given the historical observed crime matrices* $\{X_t | t = 0, 1, \cdots, n\}$, *the goal is to learn a crime predictor that can predict* $X_{n+1}$, *i.e., crime distribution at time slot* $n + 1$.

### 3.2. ST-3DNet Model

ST-3DNet is a novel deep learning model initially developed in [14] for traffic raster data prediction. Figure 1 shows the ST-3DNet structure. It primarily consists of two components, i.e., a closeness component that aims at capturing spatio-temporal features of the most recent historical data, and a weekly period component that aims to describe the periodic and trend features of traffic data.

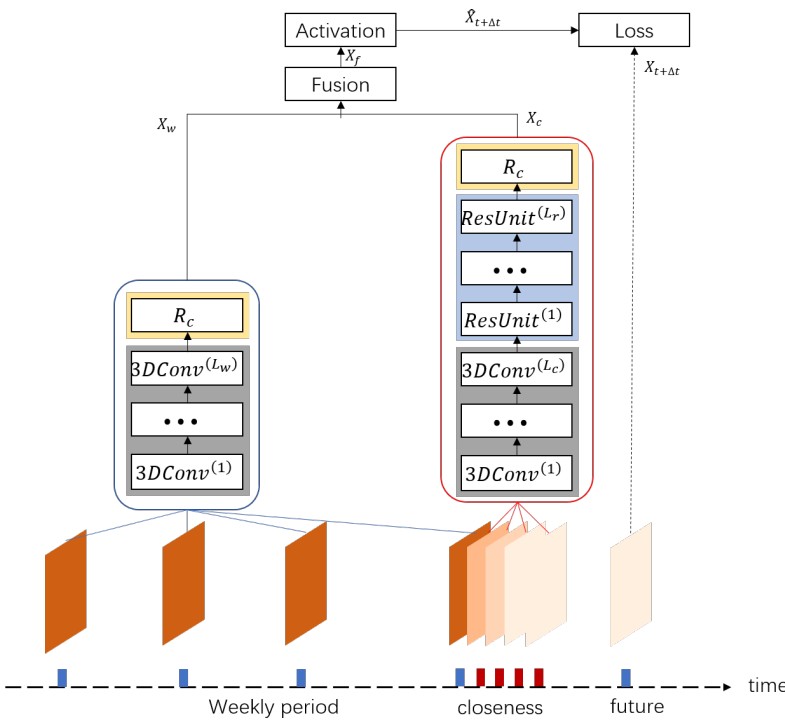

**Figure 1.** ST-3DNet structure. 3D Conv, Resunit and Rc denote 3D Convolutional, residual unit and Recalibration block, respectively.

Specifically, a studied city is partitioned into an $I \times J$ grids map by the longitude and latitude, and a grid denotes a region. For the closeness component on the right side of Figure 1, a subsequence of spatiotemporal raster data from most recent time slots serves as its input. In order to extract spatio-temporal features from traffic data, the closeness component firstly employs 3D convolution layers and 2D residual units. An Rc module is then used to distinguish and quantify the features' contribution of each region. For the weekly period component on the left side of Figure 1, the input is a subsequence of spatio-temporal data from the last few weeks. This component employs 3D convolutions to extract spatio-temporal patterns and an Rc module that selects useful features and suppresses unhelpful ones for each region. Next, the two outputs are integrated into $X_f$ according to parameter matrices that can be learned to express the contribution difference between the two temporal properties in space, shown as on the top of Figure 1. Finally, an activation function follows after $X_f$.

## 4. Methodology

### 4.1. Framework Overview

Figure 2 illustrates the framework of our proposed crime prediction model. The model inputs are the raw historical crime records, and the outputs are the hourly predicted number of crimes for all regions of a city. The proposed ST3DNetCrime includes three main

modules, namely data preprocessing, temporal features extraction, and spatiotemporal feature modeling. The three modules are described in detail in the following.

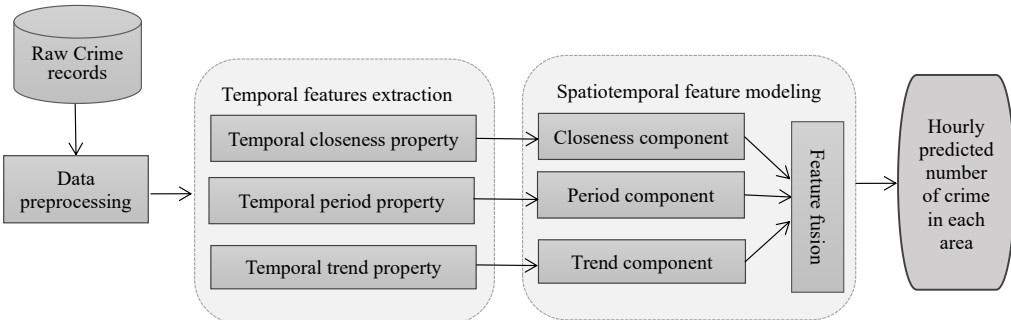

**Figure 2.** Framework.

### 4.2. Data Preprocessing

Figures 3a and 4a present crime distribution at a randomly selected time slot and the hourly crime time series over the previous two weeks in a randomly chosen grid, respectively. It shows that crime data are sparse and irregular at fine spatial temporal scales. In order to better adopt deep learning models, we use diurnal periodic integral mapping proposed in [13] to enhance the regularity of the time-series data:

$$Y_t = \int_k^t X_t \tag{1}$$

where $Y_t \in R^{16 \times 16}$ is the integrated crime matrix at the $t$-th time slot, and $k = t - (t \bmod 24)$. Figures 3b and 4b show the integrated crime distribution and the integrated hourly crime time series corresponding to Figures 3a and 4a, respectively. The integrated crime data become denser and show daily periodicity. Moreover, in order to prevent the deep learning model from abnormal training and ensure good convergence effect, the integrated crime matrix $Y_t$ is normalized through min-max normalization. The normalized integrated crime matrix is denoted by $Y_t{}^*$.

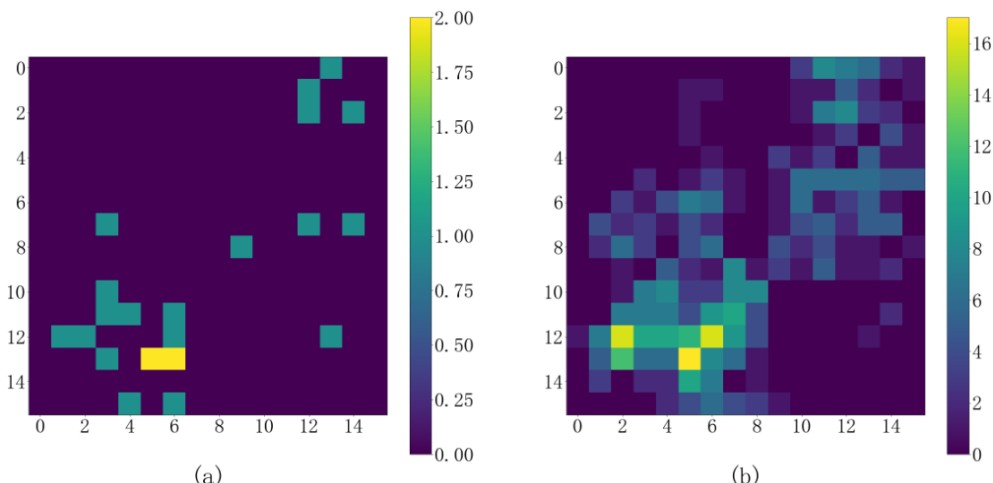

**Figure 3.** Crime distribution at a randomly selected time slot: (**a**) crime distribution at 23:00 on 8 July 2015; (**b**) integrated crime distribution corresponding to (**a**).

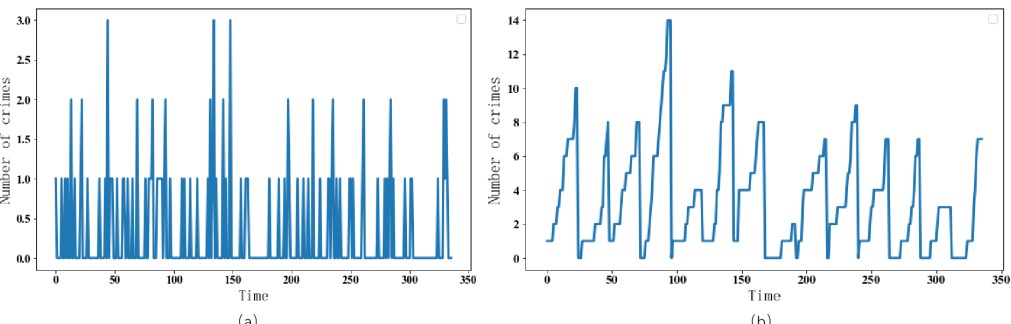

**Figure 4.** The hourly crime time series over the last two weeks in a randomly chosen grid: (**a**) hourly crime time series on the grid $[34.2000°, 34.025°] \times [-118.325°, -118.3°]$; (**b**) integrated hourly crime time series corresponding to (**a**).

### 4.3. Temporal Features Extraction

Three categories of temporal properties of crime are considered in the proposed ST3DNetCrime model, i.e., closeness, period, and trend. The three temporal properties denote the influence of recent crime data, near historical data and distant historical data on future crime, respectively. And they are formulated as follows.

$$
\begin{aligned}
Y_h &= \left\{ Y^*_{t-l_h}, Y^*_{t-(l_h-1)}, \cdots, Y^*_{t-1} \right\} \\
Y_d &= \left\{ Y^*_{t-T \times l_d}, Y^*_{t-T \times (l_d-1)}, \cdots, Y^*_{t-T} \right\} \\
Y_w &= \left\{ Y^*_{t-W \times l_p}, Y^*_{t-W \times (l_p-1)}, \cdots, Y^*_{t-W} \right\}
\end{aligned}
\tag{2}
$$

where $Y_h$, $Y_d$ and $Y_w$ are termed closeness-dependent sequence, period-dependent sequence, and trend-dependent sequence. Furthermore, $l_h$, $l_d$ and $l_w$ are the lengths of the three dependent sequences, respectively. The parameter $T$ is set to 24 because of the clear daily periodicity from Figure 4. In addition, the trend span is empirically fixed to one-week, i.e., $W = 24 \times 7 = 168$. The three dependent sequences serves as the inputs of the closeness component, the period component and the trend component, respectively.

### 4.4. Spatiotemporal Feature Modeling

Compared to the traffic data, crime data bear more complex spatiotemporal dynamic relationships, which cannot be well handled by the original ST-3DNet structure. This is mainly reflected in the following two points. Firstly, in the original ST-3DNet structure, both the periodic and trend patterns in traffic data are captured by the weekly period component. However, the periodic feature and the trend feature in crime data have different effects on crime. Therefore, it is best to design two different components to capture periodic pattern and trend pattern, respectively. Secondly, the crime within a region is not only affected by the recent crime of its nearby or distant regions, but also by the crime of its nearby or distant regions in the past. Hence, we need to consider that period and trend components can also capture spatio-temporal features. Meanwhile, the weekly period component of original ST-3DNet model only uses 3D convolutions to mainly extract periodic and trend features, and the spatial features have not been fully explored. Motivated by the above two factors, we modified the ST-3DNet structure to make it suitable for modeling spatio-temporal features in crime data, as shown in Figure 5.

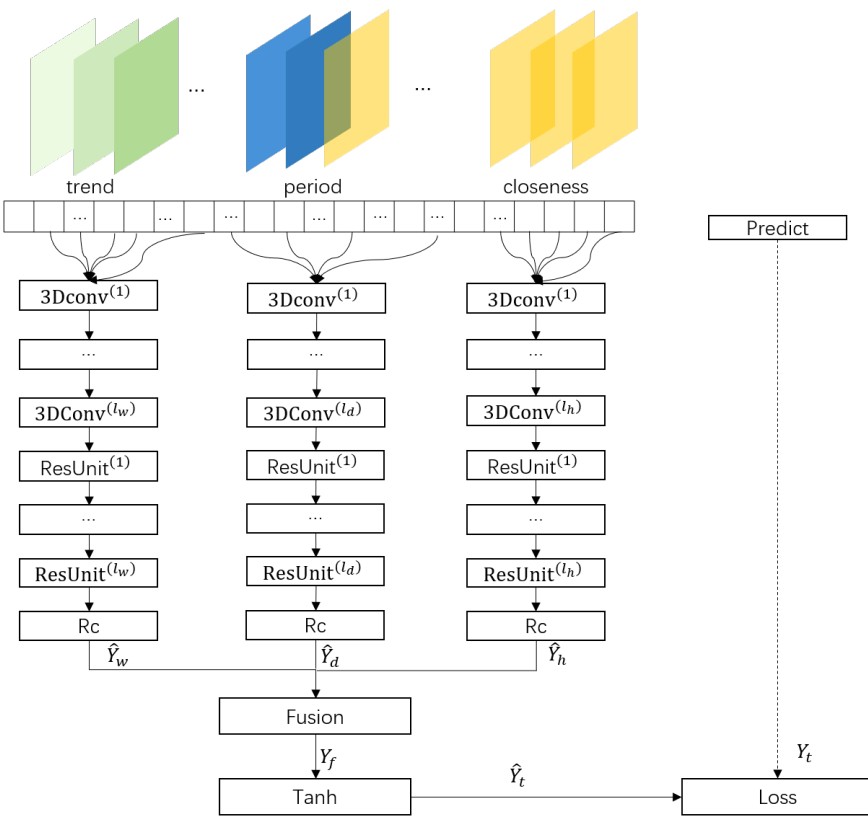

**Figure 5.** Improved ST-3DNet structure for crime prediction. 3D Conv, Resunit and Rc denote 3D Convolutional, residual unit and Recalibration block, respectively.

In the improved ST-3DNet structure, both the period component and the trend component adopt the same network structure as the closeness component of the original ST-3DNet structure. Specifically, in order to, respectively, extract spatio-temporal features from recent crime data, near historical crime data and distant historical crime data, the inputs of the three components firstly are separately fed into 3D convolution layers and 2D residual units. Next, the features' contribution of each region is distinguished and measured using an Rc module. $\hat{Y}_h$, $\hat{Y}_d$ and $\hat{Y}_w$ denote the outputs of the three components, respectively. Then, the three outputs are fused as follows.

$$Y_f = W_h \circ \hat{Y}_h + W_d \circ \hat{Y}_d + W_w \circ \hat{Y}_w \tag{3}$$

where $\circ$ is element-wise multiplication. $W_h$, $W_d$ and $W_w$ are the learnable parameters which adjust the degrees influenced by the three categories of temporal properties of crime. Finally, the predicted value denoted by $\hat{Y}_t$, is defined as

$$\hat{Y}_t = f(Y_f) \tag{4}$$

where $f$ is a nonactivation function.

## 5. Evaluation

Here, extensive experiments with real-world datasets from Los Angeles are conducted to evaluate the crime prediction performance of the ST3DNetCrime model. We mainly aim at answering the following questions:

(1) How robust is our ST3DNetCrime model? In other words, how do the different configurations of model parameters (e.g., the number of 2D residual units) affect the performance of ST3DNetCrime?

(2) How does the ST3DNetCrime perform as compared to baseline methods?

(3) How well do the ST3DNetCrime variants perform with different combinations of the three components as well as whether the data preprocessing module is used?

### 5.1. Settings

### 5.1.1. Datasets

We focus on crime prediction in Los Angeles, since it is one of the cities with the most extensive online data disclosure. This study's model for predicting crime is transferable to different urban settings. We collected crime data from 00:00, 1 July 2015 to 00:00, 1 January 2016. In total, there were 110,662 valid crime records. Each record has detailed information about the time and location (i.e., latitude and longitude). The total spatial distribution of crime is depicted in Figure 6. Geographically, these crimes occurred in the latitude and longitude ranges of $[33.7058°, 34.3343°]$ and $[-118.7668°, -118.1574°]$, respectively. From Figure 6, it is found that there are few or no crimes in a large portion of the area. Therefore, the crimes that happened within the area $[33.9000°, 34.3000°] \times [-118.6000°, -118.2000°]$ are considered in this study, shown as area Z in Figure 6. There are 98,731 crime records in the studied area, accounting for 89.2% of all crimes.

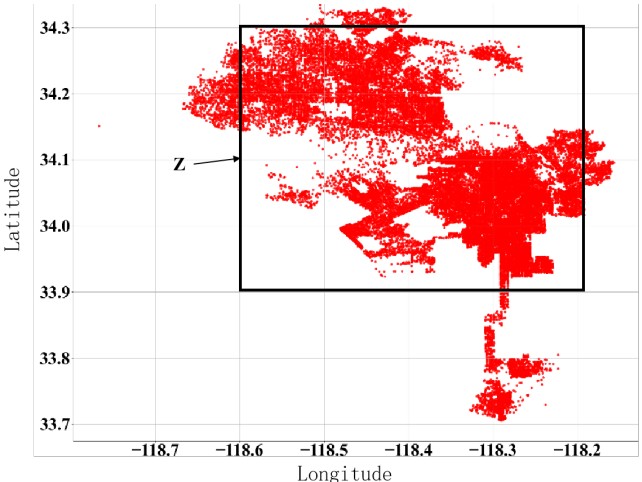

**Figure 6.** The overall spatial crime distribution.

### 5.1.2. Experimental Setup

Based on Keras, which employs Tensorflow as its backend engine, the ST3DNetCrime is implemented. The basic experimental setup is given as follows.

(1) In the crime datasets, we choose all data prior to the last two weeks as the training set, and data of the last two weeks as test set.

(2) The crime prediction performance is evaluated by the most commonly used evaluation metrics for regression problems [29], namely root mean square error (RMSE).

(3) Several hyperparameters should be preset prior to building the ST3DNetCrime model. By extensive contrast experiments, the basic structure of the ST3DNetCrime model is obtained. For each component, there are three 3D convolution layers. Filters of size $l \times 3 \times 3$ are used in the first 3D convolution layer, where $l$ is the magnitude of the input data's temporal dimension. Filters of size $3 \times 3 \times 3$ are used in the remaining two 3D convolution layers, and the 2D convolution layers in residual units employ filters of size $3 \times 3$. The model's activation functions are all ReLU. In addition, Adam with default parameters is used as the optimizer and L2-norm is used as the loss function when training the model.

### 5.2. Parameter Sensitivity Studies

To discuss the robustness of the ST3DNetCrime model, we examine how the different configurations of model parameters (i.e., the number of 2D residual units, the number of filters used in convolution layers, and the lengths of the three dependent sequences) affect

the performance of the ST3DNetCrime. In this study, the number of filters in convolution layers and the number of 2D residual units are selected from the candidate sets $\{8, 16, 32, 64\}$ and $\{2, 4, 6, 8\}$, respectively. For the lengths of the three dependent sequences (i.e., closeness-dependent sequence, period-dependent sequence, and trend-dependent sequence), we set them as $l_c \in \{2, 4, 6\}$, $l_p \in \{1, 2, 3\}$, and $l_q \in \{1, 2, 3\}$. Hence, there are 27 combinations of three independent sequence lengths. Figure 7 shows the evaluation results. We obtain two key observations from Figure 7.

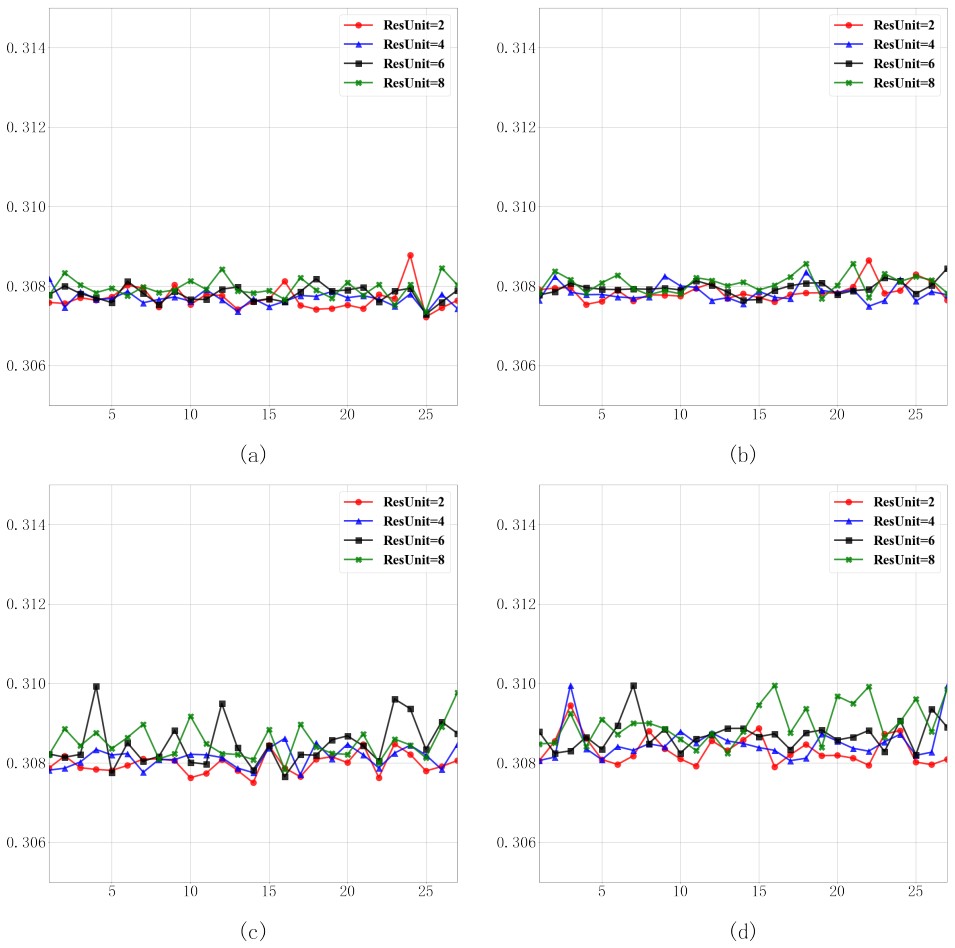

**Figure 7.** Parameter sensitivity study on the performance of ST3DNetCrime: (**a**) the number of filters in the convolution layers is 8; (**b**) the number of filters in the convolution layers is 16; (**c**) the number of filters in the convolution layers is 32; (**d**) the number of filters in the convolution layers is 64. Units: *x*-axis: number of combinations of three independent sequence lengths; *y*-axis: RMSE value.

(a) When the number of filters in convolution layers is 8 and 16, respectively, the number of 2D residual units and the lengths of the three dependent sequences have a relatively low impact on the performance of ST3DNetCrime. Furthermore, in the two scenarios, the ST3DNetCrime achieves relatively similar performance. When the number of filters in convolution layers is increased to 32, the fluctuation of RMSE is relatively large under different lengths of the three dependent sequences when the number of residual units is set to 8 and 16. If the number of filters in convolution layers is further increased to 64, the fluctuation of RMSE becomes conspicuous irrespective of what the number of residual units is set to, and the RMSE is also significantly higher than that when the number of filters in convolution layers is set to 8, 16 and 32.

(b) Overall, when the number of residual units is set to 2 and 4, the RMSE is lower than that when the number of residual units is set to 8 and 16 under most different combinations

of three independent sequence lengths. Even if the number of filters in convolution layers is set to 8 and 16, this phenomenon also exists if we look closely at Figure 7a,b.

The observations show that the proposed ST3DNetCrime has good prediction performance and strong robustness when the number of 2D residual units is small (i.e., the model structure is relatively simple) and the number of filters in the convolution layers is small (i.e., the model training time is reduced).

### 5.3. Performance Comparison for Crime Prediction

#### 5.3.1. Baselines

We compared the performance of ST3DNetCrime with the following baseline methods.

- Support vector regression (SVR) [30]: SVR is a representative machine learning method for handling regression problems. Here, a linear kernel function is used, and the time lag is chosen from a candidate sets {2, 3, 4, 5, 6, 7, 8, 9, 10, 11, 12} to find the best RMSE value.
- Long short-term memory neural network (LSTM) [31]: LSTM is a type of recurrent neural network and can learn long time-series data. A single LSTM layer is used in this case, and the number of nodes in the LSTM layer is 200. In addition, the time lag is also chosen from a candidate set {2, 3, 4, 5, 6, 7, 8, 9, 10, 11, 12}.
- Convolutional LSTM (ConvLSTM) [32,33]: ConvLSTM is a variant of the LSTM model, which changes the fully connected layer of the traditional LSTM to the convolutional layer. It is good at capturing spatio-temporal features. Two ConvLSTM layers are used here. Each ConvLSTM layer uses 8 filters with size $3 \times 3$. The time lag is also chosen from a candidate set {2, 3, 4, 5, 6, 7, 8, 9, 10, 11, 12}.
- ST-ResNet [25,26]: ST-ResNet is a deep residual-network-based prediction model for spatio-temporal data. Here, the convolutions in all residual units and Conv1 use filters with size $3 \times 3$, and Conv2 uses a convolution with one filter of size $3 \times 3$. In addition, the number of 2D residual units, the number of filters used in convolutions, and the lengths of the three dependent sequences are set the same as those of ST3DNetCrime.
- Original ST-3DNet: The weekly period component has three 3D convolution layers and its input is the same as that of the trend component in ST3DNetCrime. The other parameters are set the same as those of ST3DNetCrime.
- ST-3DNet-s: We also consider a special scenario of the original ST-3DNet, in which the weekly period component only uses one 3D convolution layer with filter of size $l \times 1 \times 1$, where $l$ is the magnitude of the input data's temporal dimension. In this case, the 3D convolution layer is used to only extract periodic and trend features along the temporal dimension.

#### 5.3.2. Comparison and Analysis of Results

Table 1 presents the RMSE of all compared methods. Overall, our ST3DNetCrime outperforms all the baseline methods. Specifically, we make the following observations.

**Table 1.** Comparison between different models in terms of RMSE.

| Model Name | RMSE |
|:---:|:---:|
| SVR | 1.323777 |
| LSTM | 0.406879 |
| ConvLSTM | 0.397032 |
| ST-ResNet | 0.312312 |
| Original ST-3DNet | 0.312203 |
| ST-3DNet-s | 0.307865 |
| ST3DNetCrime | 0.307223 |

(a) The RMSE of SVR is significantly higher than that of the deep learning methods. The reason may be that it is difficult for SVR, a traditional machine learning model, to capture the complex features from the spatial-temporal crime data.

(b) Compared with LSTM and ConvLSTM, the RMSE of our ST3DNetCrime model is reduced by 24.5% and 22.6%, respectively. As can be seen, ConvLSTM performs better than LSTM because ConvLSTM can simultaneously capture spatial and temporal features in crime data. However, ConvLSTM can only stack a few layers due to its complicated structure, thus ConvLSTM can only capture the nearby spatial feature and recent temporal feature, i.e., ConvLSTM cannot capture the spatio-temporal correlation in crime data adequately. As a result, the performance advantage of ConvLSTM is not as great as expected when compared with LSTM, while the performance of ConvLSTM is significantly worse than that of the deep residual-network-based prediction models (i.e., ST-ResNet and ST-3DNet) which can effectively model the complex correlations of spatio-temporal data.

(c) Although ST-ResNet and ST-3DNet can effectively model the complex correlations of spatio-temporal data, the RMSE of ST3DNetCrime is still reduced about by 1.63%. This illustrates that our ST3DNetCrime is more suitable for capturing spatio-temporal features in crime data and crime prediction at fine spatial temporal scales. In addition, we found an interesting special scenario of original ST-3DNet (i.e., ST-3DNet-s whose weekly period component is set to be used only for extracting periodic and trend features along the temporal dimension). In this case, although the RMSE of our ST3DNetCrime model is only slightly lower than that of ST-3DNet-s, the RMSE fluctuation of ST3DNetCrime under different parameters is much lower than that of ST-3DNet-s, as shown in Figure 8. The observations demonstrate that the proposed ST3DNetCrime framework not only reduces the RMSE of crime prediction, but also bears good robustness.

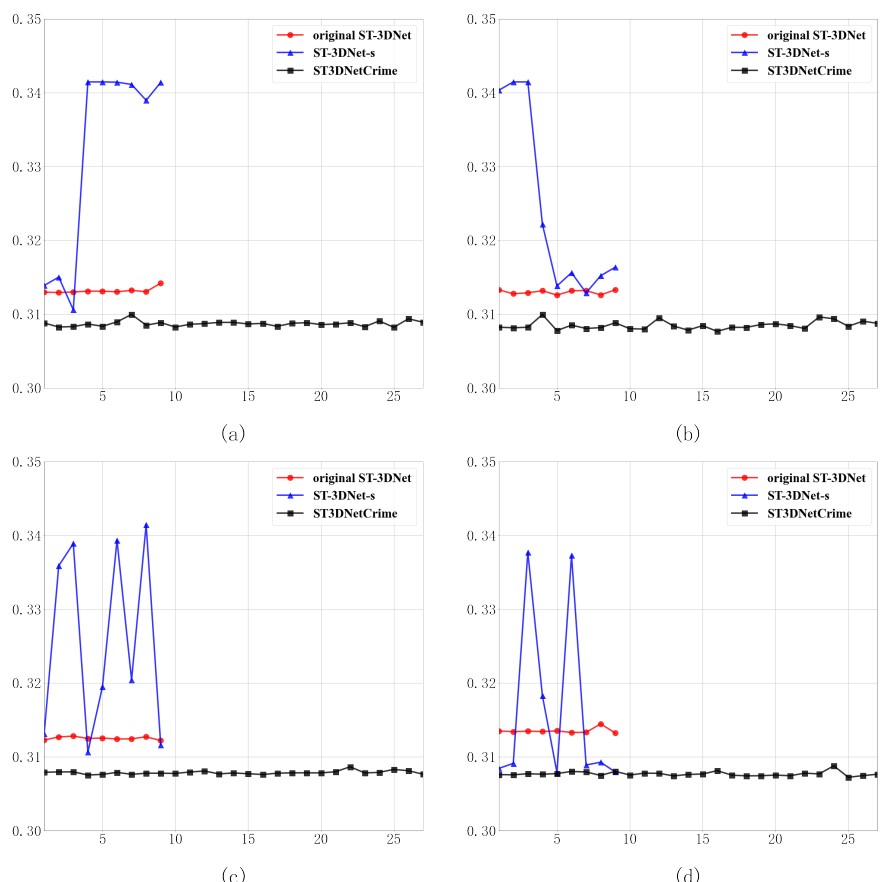

**Figure 8.** RMSE comparison under different values of $l_c$, $l_p$ and $l_q$ in four randomly selected combinations of the number of filters in the convolution layers ($N_{filters}$) and the number of residual units ($N_{res}$): (**a**) $N_{filters} = 64$, $N_{res} = 6$; (**b**) $N_{filters} = 32$, $N_{res} = 6$; (**c**) $N_{filters} = 16$, $N_{res} = 2$; (**d**) $N_{filters} = 8$, $N_{res} = 2$. Units: *x*-axis: number of combinations of the independent sequence lengths; *y*-axis: RMSE value.

*5.4. Evaluations on Variants of ST3DNetCrime*

In order to better understand the proposed ST3DNetCrime model and evaluate whether each component plays a crucial role in crime prediction, we consider four variants of the proposed model according to different combinations of the three categories of temporal properties and whether the data preprocessing module is used. The full version of ST3DNetCrime introduced in Section 4 is defined as ST3DNetCrime-f. Table 2 gives specific combination settings and corresponding model name definitions. In the evaluation, the number of 2D residual units and the number of filters in the convolution layers are fixed to 2 and 16, respectively. The evaluation results are shown in Figure 2. From the results, four key observations are summarized as follows.

(a) Compared with ST3DNetCrime-nopre, not only is the RMSE of ST3DNetCrime significantly reduced, but the RMSE fluctuation of ST3DNetCrime under different parameters is also much lower. The observation verifies that diurnal periodic integral mapping used to solve the problem of sparse and irregular in crime data not only helps to reduce the RMSE of ST3DNetCrime but also improve robustness.

(b) ST3DNetCrime-f outperforms ST3DNetCrime-cp, ST3DNetCrime-ct as well as ST3DNetCrime-pt. The RMSE value of ST3DNetCrime-f is reduced by at least 1.07% compared with the three variants. This observation suggests that spatio-temporal features, respectively, captured by closeness component, period component and trend component are all helpful to improve the prediction accuracy.

(c) Overall, ST3DNetCrime-cp performs slightly better than ST3DNetCrime-ct, and ST3DNetCrime-ct performs slightly better than that of ST3DNetCrime-pt. This indicates that the closeness component has the greatest effect on helping ST3DNetCrime make the correct prediction, while the trend component has the least impact. In other words, the more recent the crime data, the greater the impact on future crimes.

(d) The difference in prediction performance among ST3DNetCrime-cp, ST3DNetCrime-ct and ST3DNetCrime-pt is small, while the prediction performance of ST3DNetCrime-f is conspicuously better than the three variants of ST3DNetCrime. This indicates that only when the spatio-temporal features in recent crime data, near historical data and distance historical data are fully captured simultaneously, can ST3DNetCrime achieve the best crime prediction performance.

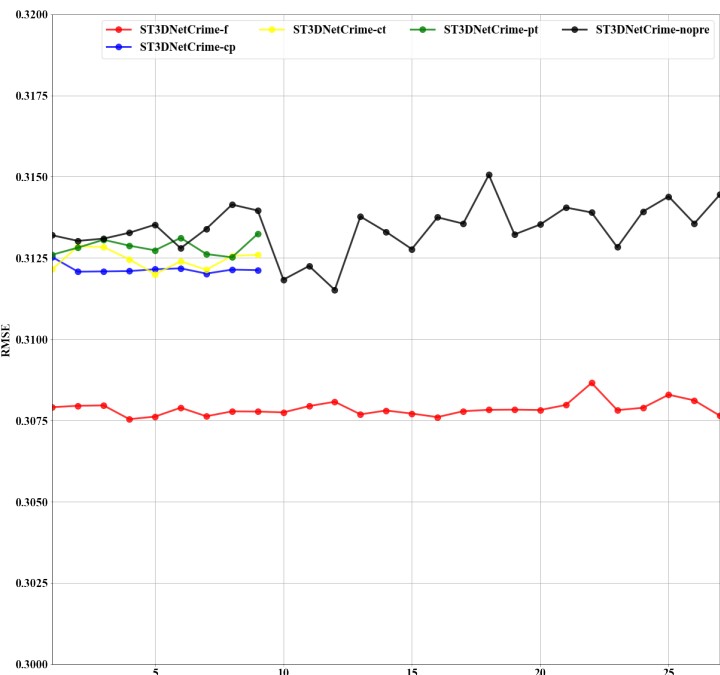

**Figure 9.** The evaluation results of variants of ST3DNetCrime. Units: *x*-axis: number of combinations of the independent sequence lengths; *y*-axis: RMSE value.

**Table 2.** Combination settings of temporal properties and corresponding model name definitions.

| Model Name | Closeness Component | Period Component | Trend Component | Data Preprocessing |
|---|---|---|---|---|
| ST3DNetCrime-f | √ | √ | √ | √ |
| ST3DNetCrime-nopre | √ | √ | √ | |
| ST3DNetCrime-cp | √ | √ | | √ |
| ST3DNetCrime-ct | √ | | √ | √ |
| ST3DNetCrime-pt | | √ | √ | √ |

## 6. Conclusions

In this paper, we proposed the ST3DNetCrime model for crime prediction at small spatial and hourly temporal scales, which mainly comprise a closeness component, period component and trend component, respectively, designed to capture spatio-temporal features of recent crime data, near historical crime data and distant historical crime data. We evaluate the designed model through extensive experiments with real-world datasets from Los Angeles. The results showed that ST3DNetCrime has good prediction performance and strong robustness especially when the number of 2D residual units is small (i.e., the model structure is relatively simple) and the number of filters in the convolution layers is small (i.e., the model training time is reduced). Compared with the baseline methods, ST3DNetCrime not only reduces the RMSE, but also has good robustness. In addition, we discovered that only when the spatio-temporal features in recent crime data, near historical crime data and distance historical crime data are fully captured simultaneously, can ST3DNetCrime achieve the best crime prediction performance.

In the future, there are some research areas worth exploring. First, we would like to improve the proposed ST3DNetCrime model by using crime-related auxiliary data, such as weather data, holiday data, points-of-interest (POI), and public service complaints. Second, we believe that it is interesting but challenging to study crime prediction at fine spatial temporal scales in the context of regular anti-COVID-19 management. Third, we did not distinguish between crime types. In fact, the spatial-temporal patterns behind different types of crime are vary widely, and the crime data are more sparse. Hence, although a challenging feat there is value in considering models for different crime types separately. Finally, we need to consider adding a trigger [34,35] to the model to determine in what intervals the deep learning model should be rebuilt to maintain the accuracy of crime prediction.

**Author Contributions:** Conceptualization, Qifen Dong and Yu Li; methodology, Qifen Dong, Yu Li, and Ziwan Zheng; software, Yu Li; validation, Qifen Dong and Yu Li; investigation, Ziwan Zheng; data curation, Yu Li; writing—original draft preparation, Qifen Dong; writing—review and editing, Xun Wang and Guojun Li; visualization, Xun Wang; supervision, Qifen Dong; project administration, Qifen Dong; funding acquisition, Qifen Dong. All authors have read and agreed to the published version of the manuscript.

**Funding:** This research was supported by Zhejiang Provincial Natural Science Foundation of China under Grant No. LQ18G010001 and National Natural Science Foundation of China under Grant No. 41901160.

**Institutional Review Board Statement:** Not applicable.

**Informed Consent Statement:** Not applicable.

**Data Availability Statement:** Not applicable.

**Acknowledgments:** We thank the anonymous reviewers and academic editor for the constructive suggestions and insightful comments which substantially improved the quality of this paper.

**Conflicts of Interest:** The authors declare no conflict of interest.

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
