# Peer review of "ST3DNetCrime: Improved ST-3DNet Model for Crime Prediction at Fine Spatial Temporal Scales"

_ijgi, doi:10.3390/ijgi11100529_

Round 1
Reviewer 1 Report
The authors developed an improved ST-3DNet structure for crime predicting at finer spatio-temporal scales, which promotes the accuracy of crime prediction.
There are still some advices:
1. In Line 127-131, the authors mentioned that diurnal periodic integral mapping method was applid to the time series data. Does the original data and processed data have different prediction accuracy for the model? Why this procedure important in the data preprocessing?
2. The authors used crime data in Los Angeles, what's the crime type in the data? Does different crime type of the data all perform well in the ST-3DNetCrime model?
3. Some wrong spelling exists. For example, in line 198 "read-world" should be "real-world", etc. The authors should go through the whole manuscript carefully.
4. The authors only compared results with ST-3DNet-based machine learning models. But whether the model performed betther than other machine learning models, such as random forest, svm, LSTM, is unknow. The authors should compared the ST-3DNetCrime model with broader existed models.
Reviewer 2 Report
This paper presents an interesting model for predicting crime. The paper is well written and on the whole adds value to the field.
1) One area that could be addressed is justification of the measure used to validate the model (RMSE). Why was this adapted. There are other measures used - for example Prediction Accuracy Index, or the Mean Absolute Error. The paper would be strengthened by justifying your selection.
2) I draw your attention to a recent systematic review of crime forecasting to support your literature. https://doi.org/10.1186/s40163-020-00116-7
Please comment on
2.1) how your model fits in with previous studies and how this adds value - are the categorisations of models you discuss consistent in your literature review consistent with this systematic review
2.2) the metrics used for forecasting (linked to comment 1)
2.3) how does your model address prediction error across space, and also issues such as over-fitting, multi-collinearity, sampling bias, and data sparsity.
3) Is this model for all crime types. There is value in considering models for different crime types separately (eg burglary versus violence) as spatial-temporal patterns will be very different- based on theories you identify in your literature (routine activity, rational choice etc). This may be a future consideration although examining by different crime types would reduce your sample size.
Reviewer 3 Report
No need to indicate the evaluation method in the abstract especially when it requires the acronym.
"mean square error(RMSE) is used to evaluate the crime prediction performance"
the abstract is not clear. mostly the authors must elucidate the problem and the scientific contribution of the proposed method.
The introduction must elucidate the problem, challenges, and novel contributions of the proposed method. However, referring to the reference [3-11], it seems like literature that is also quite old, citing the works from 20 years ago till almost 5 years ago which is not necessary to discuss these old approaches in crime prediction.
Even though the authors must explain which problem they solve, they are mostly focused on the application: Crime prediction.
The authors must clearly explain why the problem of space-time series prediction is important and how they have solved this problem in a novel manner.
The novelty of the approach is not clear. There are only two bullet points for contributions in the introduction section where the second is for the experiment and the first one does not indicate the main novelty of the proposed solution.
The authors must separate the related work section from the introduction. Add a new subsection for the related work section after the introduction.
Missing references:
1. Najafi et al.: SoulMate: Short-text author linking through Multi-aspect temporal-textual embedding
The time of the hour is also affected by the day on which the crime happens. So the time is affected by the Temporal Subset Property. you need to report this in the related work, even though you only consider the Hour granularity.
2. Saeid et al. Mining subgraphs from propagation networks through temporal dynamic analysis.
Given your model, the trigger should determine in which intervals the deep learning model should be rebuilt. so you need to indicate the concept of triggering which necessitates the model to rebuild itself to maintain the accuracy in the prediction of the crimes.
Dataset description is related to the experiments not to come before methodology.
some parts of the paper are naive and they are not required to be over-explained. For example, feature scaling is common knowledge, referring to Eq. 2.
While figures 4 and 5 depict a deep learning structure, I would suggest that you alternatively add a framework overview section that can show the overall structure of your proposed method. You will need to separate the offline and online components to better reveal the time complexity of your model.
A framework overview can establish the main scientific contributions of the article.
Is RMSE adequate to correctly reflect the accuracy of the baselines?
Moreover, the authors must add a section for the baselines and then compare their method with another rival, preferably with trending state-of-the-art models.
Round 2
Reviewer 3 Report
Even though the background and research design can be improved, and the research is of an average rank, however, the authors have addressed the concerns. Hence, I propose the work to get published (Accepted). With a hope that their future research will be even better than the current via using the concept of triggering in model rejuvenation.